# Single-Molecule Analysis of the Improved Variants of the G-Quadruplex Recognition Protein G4P

**DOI:** 10.3390/ijms241210274

**Published:** 2023-06-17

**Authors:** Paras Gaur, Fletcher E. Bain, Masayoshi Honda, Sophie L. Granger, Maria Spies

**Affiliations:** Department of Biochemistry and Molecular Biology, Carver College of Medicine, University of Iowa, Iowa City, IA 52242, USAmasayoshi-honda@uiowa.edu (M.H.);

**Keywords:** G-quadruplex (G4), G4-recognition, G4P, single-molecule total internal reflection fluorescence microscopy (TIRFM)

## Abstract

As many as 700,000 unique sequences in the human genome are predicted to fold into G-quadruplexes (G4s), non-canonical structures formed by Hoogsteen guanine–guanine pairing within G-rich nucleic acids. G4s play both physiological and pathological roles in many vital cellular processes including DNA replication, DNA repair and RNA transcription. Several reagents have been developed to visualize G4s in vitro and in cells. Recently, Zhen et al. synthesized a small protein G4P based on the G4 recognition motif from RHAU (DHX36) helicase (RHAU specific motif, RSM). G4P was reported to bind the G4 structures in cells and in vitro, and to display better selectivity toward G4s than the previously published BG4 antibody. To get insight into G4P- G4 interaction kinetics and selectivity, we purified G4P and its expanded variants, and analyzed their G4 binding using single-molecule total internal reflection fluorescence microscopy and mass photometry. We found that G4P binds to various G4s with affinities defined mostly by the association rate. Doubling the number of the RSM units in the G4P increases the protein’s affinity for telomeric G4s and its ability to interact with sequences folding into multiple G4s.

## 1. Introduction

Single-stranded nucleic acids often fold into non-canonical structures [1,2]. Among these, guanine-rich sequences possess the capacity to fold into four-stranded arrangements known as G-quadruplexes (G4s) which are stabilized by Hoogsteen hydrogen bonding and monovalent cations [3]. Sequences that have four or more guanine tracts can form intramolecular G4s with various topologies [4]. There are nearly a million distinct sequences in the human genome that were predicted to form intramolecular G4s [5,6], and about 10,000 of these G4s were shown to form in chromatin of human cells [7,8]. G4s play important physiological and pathological roles in telomere biology, DNA replication, repair, epigenetics, cell differentiation, gene regulation and as sensors of oxidative stress [9,10,11,12,13,14,15,16,17,18,19]. Due to the difference in length and sequence composition of G4-forming sequences, G4s may fold into structures with different topologies and stabilities. Some G4-forming sequences can assume different topologies depending on solution conditions, crowding, temperature, adjacent sequences and presence of ligands. Human telomeric G4 (hTelG4), for example, can fold into several topologically distinct conformations that interconvert [4,20,21]. NMR and X-ray crystallography studies have observed hTelG4 in parallel configuration [22], antiparallel configuration with a loop placed diagonally across a terminal G-tetrad [23,24] and hybrid configurations [24]. 

The G4 structures in DNA and RNA play multiples roles in cancer development and progression [25], and resolution of G4s in DNA during DNA replication, RNA transcription and DNA repair is important for genome integrity [9]. The regulatory and promoter regions of many genes associated with the major hallmarks of cancer, such as sustained angiogenesis, uncontrolled DNA replication, apoptosis evasion, self-sufficiency, and metastasis contain G4-forming sequences. Examples of such genes include *MYC*, *VEGF*, *KIT* and *BCL2*, among others. The proto-oncogene *MYC*, which is overexpressed in the majority of solid tumors, is a well-studied example of a gene with a G4 forming sequence in its promoter region [26,27,28,29,30]. The functional importance of G4-mediated regulation of expression of oncogenes and tumor suppressors in different cancers has established them as a desirable target for therapeutic interventions. One popular approach is to target G4 DNA through the use of small molecule ligands, which can stabilize these structures and induce DNA damage for therapeutic purposes. A database comprising over 3200 reported G4 DNA ligands has been documented in several studies [31,32]. Some of the commonly used ligands include Telomestatin, which targets telomeric G4 repeats and inhibits telomerase activity by disassembling shelterin proteins; Pyridostatin, another G4 ligand that stabilizes G4 structures, and TMPpyP4, which stabilizes G4 sequences in the promoter region of *MYC* gene, downregulating its expression [33,34,35,36,37,38]. 

Some proteins interact with specific subsets of G4 structures, while others are less specific to the G4 fold [39,40]. An important milestone in the field was the selection of BG4 antibody which has become a valuable tool for visualization of G4 in mammalian cells [28]. To improve on the BG4, a small-artificial protein called G4P was constructed based on the G4 recognition motif from RHAU (also known as DHX36 or G4R1) helicase. G4P was shown to bind G4 structures both in vitro and in cells and demonstrated improved selectivity toward G4s when compared to the BG4 antibody [41]. However, both BG4 and G4P bind hTelG4 quite poorly. RHAU belongs to the DEAD/H-Box family of RNA helicases [42], and is known to exhibit a marked preference for parallel G4s [43,44]. RHAU is a 1008-amino acid protein consisting of 3 regions: the N-terminal region, the helicase core region and the C-terminal region. The N-terminus of RHAU (1-203 aa) contains a glycine-rich region and the RHAU-specific motif (RSM), both of which are important for recognizing and resolving G4 DNA structures. Multiple sequence alignment of the RSM region of RHAU helicases has shown that RSM has a conserved 13-amino acid sequence involved in recognizing G4 structures [45]. The first structural information on the RHAU G4 binding came from the NMR structure of the RHAU RSM peptide bound to a parallel G4 (T95-2T), which contained 4 runs of three G’s separated by loops composed of 1 T, and 2 Ts at the 5′ end of the G4-forming sequence (PDB 2N21; Appendix A) [43]. Part of the RSM peptide in this structure formed an α-helix with the rest of the peptide being disordered. The same peptide formed a more extended α-helix when present as a part of RHAU protein bound to *c-MYC*G4 (PDB 5VHE; ref. [46]). G4P contains two RSM peptides connected by a flexible linker potentially allowing G4P to bind to two surfaces of the G4 structure [41]. Disorder probability prediction suggests that both RSM peptides in G4P are structured with the linker regions being disordered (Appendix A). Specificity of G4P for a particular type of G4 can be caused by accessibility of the terminal G-tetrads as well as by the compatibility between the size of the quadruplex and the length of the linker.

RSM-like motif has also been identified in human FANCJ helicase [47]. In contrast to RHAU, however, FANCJ readily interacts with hTelG4 in its antiparallel/hybrid conformation. FANCJ both unfolds and refolds G4s, and the process of refolding of the hTelG4 results in appearance of the parallel conformation [47]. We reasoned, therefore, that a FANCJ-derived RSM may improve G4P binding to telomeric G4s.

The goal of this study was to develop better protein tools for visualizing G4 structures in vitro and in living cells, and to be able to visualize G4s irrespective of their topology and without affecting their thermodynamic stability. This objective is distinct from small molecule discovery studies that attempt to develop compounds that selectively bind specific G4s. In this study, we constructed and evaluated several G4P derivatives, including G4Px2 (2G4P), FJG4P and FJG4Px2 (2FJG4P), to enhance the stability of G4P-G4 complexes. The G4 binding motif from human FANCJ helicase, which is known for its strong affinity toward telomeric G4, was incorporated into the latter constructs. Understanding the nature of the selective binding to different types of G4s and development of better, less selective reagents can be greatly assisted by kinetic analyses of the sensor–G4 interactions. To this end, we applied single-molecule total internal fluorescence microscopy (smTIRFM) and mass photometry (MP) to probe the association kinetics of our G4P variants with several types of G4s, as well as the stabilities and stoichiometries of the resulting complexes. 

## 2. Results

### 2.1. Affinity of the G4P Protein for G-Quadruplexes Can Be Improved by Increasing the Number of G4-Recognition Elements

The original G4P protein contains two RSM peptides from RHAU helicase, which function as G4 sensors (Figure 1A) [41]. Previously, we found that the amino acid sequence surrounding K141 and K142 (129-PEKTTLAAKLSAKKQ-143) in human FANCJ helicase is reminiscent of RSM and shares the “AKKQ” anchoring motif (Figure 1A) [47]. Based on these observations, we proposed that proline 129 in FANCJ acts as the initiation site for an α-helix formation that interacts with G4 DNA through K131, K137 and K141/142, similar to the mechanism proposed for RHAU18 peptide based on the NMR structure [43]. To determine if FANCJ derived RSM is less selective than RHAU-derived sequence, we developed FJG4P. We have also doubled the number of RSMs by constructing 2G4P and 2FJG4P proteins (Figure 1; the complete protein sequences are provided in Appendix A). The expected binding models of the various G4Ps utilized in this investigation are depicted in Figure 1B–D. There are two mechanisms by which the original G4P protein may display a preference for parallel G4s, such as *c-MYC*. The first is complementarity between the RSM motif and the surface of the terminal tetrad, and the second is the distance between the two tetrads. Therefore, we purified four G4P variant proteins and fluorescently labeled them with Cy3 dye at the N-terminus for single-molecule analysis (Figure 1E,F). The labeling efficiencies of G4P, 2G4P, FJG4P and 2FJG4P were 55%, 71%, 42% and 28.5%, respectively.

The Electrophoretic Mobility Shift Assays (EMSAs) were performed using unlabeled G4P, which confirmed that G4P exhibits a preference for G4 DNA structures with a parallel topology. The K_d_ value calculated for the binding of G4P to *c-MYC*G4 (parallel topology, see Appendix A for sequence selection), was 84 ± 8 nM, which is about 7-fold lower than the value of 558 ± 156.8 nM calculated for human telomeric hTelG4 (mixed topology). This difference in observed affinity is consistent with previously published data, where the Kd values for *c-MYC*G4 and hTelG4 were calculated to be 2.7 nM and 14 nM, respectively, showing a 5-fold change [41]. Although there are differences in the exact values, the overall trend is comparable (Figure 2A,C and Appendix A).

We observed that increasing the number of G4 sensing elements in 2G4P construct notably enhanced the binding affinity of the protein for the G4-DNA. Notably, in EMSA experiments we observed not only improved Kds, but also formation of slower migrating species at higher protein concentrations, indicative of higher molecular weight species containing multiple proteins bound to the same G4 or networks of the 2G4P/G4s. We observed a 3-fold increase in affinity for the 2*G4P-c-MYC*G4 interaction compared to the *G4P-c-MYC*G4 interaction, while the 2G4P-hTelG4 interaction showed a 6.5-fold improvement over the G4P-hTelG4 interaction (Figure 2 and Appendix A). 

Because the full-length FANCJ helicase readily binds human telomeric G4, we expected FJG4P and 2FJG4P to bind hTelG4 and *c-MYC*G4 with similar affinities. However, FJG4P and 2FJG4P exhibited weaker binding towards both *c-MYC*G4 and hTelG4 compared to G4P. In EMSA experiments, we did not observe any improvement in binding to *c-MYC*G4 when using FJG4P. FJG4P showed no detectable binding to hTelG4, but its binding improved significantly in the 2FJG4P, resulting in a Kd value of approximately 4.5 µM (Figure 2, Appendix A). 

### 2.2. Association Kinetics Plays an Important Role in the G4P-hTelG4 Complex Formation

To evaluate *G4P-c-MYC*G4 and G4P-hTelG4 binding kinetics, we used smTIRFM. In the initial smTIRFM experiments, the pre-folded biotinylated G4 DNA was tethered to the surface of the reaction chamber, and the Cy3-labled G4P protein was infused into the chamber. Fluorescence trajectories, which show the time-based changes in Cy3 fluorescence in a specific location in the TIRFM reaction chamber, represent Cy3-G4P binding to and dissociation from individual surface-tethered G4 molecules. Figure 3A–D and Appendix A show representative trajectories and their analysis with raw fluorescence data shown in green, overlaid with an idealized trajectory represented by a black line. We observed dynamic binding and dissociation of monomers of G4P to/from each quadruplex with dwell times lasting a few seconds (see analysis in Figure 3 and Table 1). No binding was observed in control experiments that had no surface-tethered DNA. The binding of monomers was inferred from a single-step change in the fluorescence. All trajectories in each experiment were collectively analyzed using hFRET, a robust program that enables state selection in the presence of heterogeneity by using the variational approximation for Bayesian inference to estimate the parameters of a hierarchical hidden Markov model [48]. Most data were best described by the two-state model (bound and free), except the 2G4P and 2hTelG4 (2 telomeric repeats of G4 DNA), which showed 3 states (see Appendix A). The dwell times were then extracted and sorted using KERA [49]. The 2 states are labeled as unbound/state 1 and bound/state 2. The dwell time is represented by τ, and we obtained values for both the “ON” dwell time (τ_on_) and “OFF” dwell time (τ_off_) for each data set. Using a single-exponential function and considering durations of all bound events binned in different time intervals, we obtained a dissociation rate constant k_off_, which was independent of the protein concentration. The dwell-time distribution was constructed from all unbound data and fitted with a single-exponential function, yielding the association rate v_+1_, which increases with increasing protein concentration. For all data sets, the association rate constants (k_on_) were calculated from respective v_+1_ values, and protein concentration adjusted by labeling efficiency. The equilibrium dissociation constant was calculated from the rate constants as K_d_ = k_off_/k_on_ (Table 1 and Appendix A). It is worth pointing out that we observed robust, albeit not very stable, binding for all constructs and substrates. The affinities calculated from the single-molecule experiments were significantly higher than those calculated from EMSAs, reflecting a non-equilibrium nature of gel-based binding experiments that commonly underestimate binding of rapidly associating and dissociating species.

Overall, increasing the number of RSM elements—and thereby, increasing the number of contacts with G4—increased the binding affinity. In the case of EMSA, assays binding to *c-MYC*G4 improved 2.5-fold from Kds of 84.14 ± 8 nM for G4P to 27.66 ± 4 nM for 2G4P, while in smTIRF-based experiments it improved from 0.22 ± 0.02 to 0.16 ± 0.01 nM, which is a 1.5-fold change. Notably, this increase in affinity was due to an increase in the association rate constant as the dwell time of bound state (τ ≈ 1.6 s) remained the same for the two proteins. In the case of G4P-hTelG4 and 2G4P-hTelG4, we observed a ~3-fold increase in the dwell time of the bound state, which was 1.5 ± 0.03 s for the former and improved to 3.5 ± 0.2 s, consistent with an increase in affinity. Another significant increase in the dwell time of bound state was observed in the case of G4P binding to the substrate capable of forming two telomeric G4s, 2hTelG4, which was 2.5 ± 0.1 s, a 1.7-fold increase compared to G4P-hTelG4. Surprisingly, the association rate constant for 2G4P binding to hTelG4 was 2.3-fold slower than that of G4P binding to the same substrate (Table 1).

In the case of FJG4P-hTelG4 and 2FJG4P-hTelG4, we observed similar dwell times of 3.1 ± 0.1 s and 2.8 ± 0.06 s, respectively, but there was a 3-fold improvement in the Kd value, from 32.7 ± 4.8 to 10.7 ± 1.7, due to an increase in the association rate constant (Table 1). Notably, the dwell times of the FJG4P and 2FJG4P were approximately 3-fold longer on the hTelG4 DNA compared to *c-MYC*G4 DNA.

To confirm that surface-tethering does not influence the G4P-G4 association, we performed complementary experiments where we immobilized the proteins and infused the Cy5-labeled DNA molecules. In these experiments, we observed similar trajectories and overall binding trends (see trajectories in Appendix A).

### 2.3. G4P Binding Does Not Interfere with the G4 Structure

So far, our single-molecule experiments have shown that G4P and its variants have a strong affinity for the G4 DNA structure and that the differences in affinities between different G4s are driven by the association rate. These differences in association rates may stem from the acquisition of the G4-recognition conformation in the RSMs, or by changes in the G4 structures. Previously, FANCJ was shown to remodel telomeric G4 structure [47]. A FANCJ-derived G4-recognition peptide may potentially do the same. To determine whether G4P binding induces structural changes in the G4 DNA structure, we utilized single-molecule Förster resonance energy transfer (smFRET). 

Here, we surface-immobilized the DNA substrate with Cy3 (FRET donor) and Cy5 (FRET acceptor) dyes (as shown in the Appendix A). FRET between Cy3 and Cy5 reports on the folding state of G4, while variations in the FRET value reveal the dynamics of the system. We then introduced G4Ps and FJG4Ps into the reaction chamber at high concentrations to ensure that there is sufficient protein to bind and induce conformational changes in the G4 DNA structure, if any. We observed no changes in FRET when we introduced the G4P or G4P variants into the reaction chamber. The FRET value remained the same as the control (DNA only), indicating that the binding of G4P or G4P variants does not induce any conformational changes in the G4 quadruplexes (Appendix A). 

### 2.4. Increasing the Number of G4s Increases Valency of the Interactions

At telomeres, multiple G4 structures can arise in proximity. Multivalent interactions between such telomeric G4s and expanded versions of G4Ps may form complex interaction networks. We applied mass photometry (MP), a single-molecule technique that uses interferometric light scattering to accurately measure masses of macromolecules and macromolecular complexes in solution [50,51]. While G4P and FJG4P are below the detection limit of MP due to their small size (7.09 kDa and 9.93 kDa, respectively), 2G4P and 2FJG4P were readily detectable (17.32 kDa and 18.4 kDa, respectively; Figure 4A,E). Notably, both proteins were observed binding to and dissociating from the glass surface yielding positive and negative molecular weight distributions that mirrored each other. For both proteins, we observed formation of multimers whose molecular weight increased with increasing protein concentration (Figure 4C,D,F–H). In the presence of hTelG4, 2hTelG4 and *c-MYC*G4 DNA, both 2G4P and 2FJG4P formed higher molecular weight complexes, suggesting binding of these proteins to G4s and forming networks. The largest network size was observed in the case of 2FJG4P binding to 2hTelG4 (Figure 4G).

To confirm the network formation observed in MP data, we carried out additional smTIRF experiments. We constructed a high-density G4 lawn by immobilizing 10 nM G4 DNA substrate to ensure that the neutravidin molecules are fully saturated with biotinylated DNA (the actual density here is defined by the fraction of biotinylated PEG) and introduced to it various concentrations of G4P. The resulting trajectories are shown in Appendix A. We observed continuous traces with high fluorescence intensity (measured in AU), indicating that G4P does not fully dissociate from the DNA and continues to interact with different G4 molecules on the slide. Both the MP experiments, and the high-density single-molecule experiments suggest that increasing the number of G4s leads to an increase in the valency of these interactions.

## 3. Discussion

The human genome contains numerous G-rich sequences that can fold into G-quadruplex structures (G4s). Multiple studies have indicated that G4s play a role in regulating various cellular processes ranging from gene expression to telomere biology. G4s are frequently found in the regulatory regions of oncogenes and tumor suppressors (26–30). These structures, however, may interfere with fidelity of DNA replication and repair. The formation and processing of telomeric G4s is critical for telomere homeostasis and maintenance of the telomere length and prevention of DNA damage signaling at the ends of linear chromosomes. It should be noted that formation of such structures was observed in single-molecule experiments, and it was proposed that multiple G4s at telomeres modulate accessibility of telomeric ends [52].

Imaging of G4s is limited by the availability of good and reliable reagents that can recognize and stably bind to G4 DNA. None of the existing G4 reagents have been characterized at the single-molecule level. Previously, an engineered single-chain antibody called BG4 was reported from the Subramanian lab [53]. BG4 antibody is widely used for imaging in human cells and sequencing studies [25,53,54,55]. However, it has limitations, such as some level of cross reactivity with other types of DNA structures [56] and a selective preference for parallel G4 topology. To overcome these issues, G4P was originally constructed [41], and we analyzed G4P and its variants in this study. Here, we purified four G4P variants and analyzed their binding to G-quadruplexes with parallel and anti-parallel/mixed topologies. As previously reported, G4P binds parallel *c-MYC*G4 with high affinity but displays lower affinity for a mixed geometry telomeric G4 DNA. Unexpectedly, this difference in affinity is not a reflection of an unstable G4P-hTelG4 complex, as our single-molecule experiments showed the same dissociation rate constants for both quadruplexes (Table 1). Instead, the association rate constant was slower for hTelG4 resulting in overall lower affinity. 

To identify a better G4 DNA binding protein, we constructed and tested tandem repeats of G4P protein, referred to here as 2G4P, and a similar peptide sequence from FANCJ helicase which we named FJG4P and 2FJG4P. Our results show that tandem repeats of the G4P protein outperform single G4P for binding to both *c-MYC*G4 (parallel G4) and hTelG4 (anti-parallel/mixed G4), and exhibit a preference for parallel topology over anti-parallel or mixed. Again, an approximately 2-fold increase in the affinity of 2G4P over G4P for *c-MYC*G4 was due to an increase in the association rate constant. Surprisingly, no corresponding increase in the affinity of 2G4P for hTelG4 was observed in the single-molecule experiments despite a 2-fold increase in dwell times. 

Notably, binding of all constructs was improved by increasing the number of telomeric repeats to allow formation of two hTelG4s. Moreover, single-molecule fluorescence trajectories of the 2G4P protein binding to 2hTelG4 best fitted to a 3-state model indicating a more complex interaction comprising 2 bound states, which may arise due to G4P binding to 1 or both G4 folds of the hTelG4 DNA and subsequent detachment from the DNA molecule. Formation of the complexes between multiple G4P constructs and multiple G4s was also evident in the mass photometry experiments (Figure 4) and in a smTIRFM experiment in which G4 was tethered to the surface at high density (Appendix A). The networks were larger in the case of FJG4P constructs despite lower overall affinity for G4s. Such a propensity to form complex protein-G4 networks may become beneficial in visualization of human telomeres where multiple G4s may arise simultaneously. While the tandem repeat variants 2G4P and 2FJG4P clearly performed better than G4P and FJG4P, and can be used in the future for in vitro and in vivo studies of G4 biology, their performance can be further improved by enhancing their G4 association kinetics. This can perhaps be achieved by improving the stability of the G4-binding helices.

Recently, the Subramanian lab reported another molecule called SG4, which is a camelid heavy chain-only derived antibody [45]. SG4 displayed improved binding to various G4s. In the future, it would be interesting to do a comparative single-molecule study of SG4 antibody with G4Ps and characterize its binding kinetics.

## 4. Materials and Methods

### 4.1. Construction of Plasmids

The plasmids for the expression of codon optimized versions of G4Ps and variants with N-terminal FLAG-His tags in the pET28a backbone were synthesized by GenScript. 

### 4.2. DNA Oligos

The labeled and unlabeled DNA oligos were ordered from IDT and are listed in Appendix A. There are a few different versions of *c-MYC*G4 that have been used in literature. Sequence alignment of these sequences is shown vis-à-vis *c-MYC*G4 used in this study in Appendix A [41,57,58,59].

### 4.3. Purification and Fluorescent Labeling of G4P, 2G4P, FJG4P and 2FJG4P

*E. coli* BL21 DE3 strain was transformed with pET28b plasmid harboring G4P, 2G4P, FJG4P and 2FJG4P coding sequence with N-terminal FLAG-6xHis tag combination. The colonies were picked and grown in an LB medium supplemented with 0.05 mg/mL kanamycin at 37 °C until the OD reached 0.6–0.7. Protein synthesis was then induced by the addition of 1 mM IPTG (Isopropyl β-D-1-thiogalactopyranoside). After 4 h at 37 °C, cells were harvested by centrifugation at 4000 rpm. Cell lysates were prepared in H-500 buffer containing 25 mM Hepes at pH 7.4, 500 mM NaCl, 20 mM imidazole, 10% glycerol, 0.5 mg/mL lysozyme, 1% Triton-X, 5 mM β-mercaptoethanol and 1 protease inhibitor tablet per 10 mL (cOmplete Mini, EDTA-free protease inhibitor cocktail tablets, #11836170001, Millipore Sigma, Burlington, MA, USA). Clarified cell lysate was loaded onto a chelating HP column (GE Healthcare, Chicago, IL, USA) charged with Ni^2+^, and washed using H500 and H150 buffers until the absorbance at 280 nm reached a base line (H150 is same as H500 except the NaCl concentration which is 150 mM). The G4Ps were then eluted from the Nickel column with H150 buffer containing 300 mM final imidazole concentration. The elution fractions were immediately applied to a HiTrap Heparin HP column (GE Healthcare). The gradient from 0 to 2M NaCl was used to elute G4P from the Heparin column. The fractions containing G4P were pooled and dialyzed overnight before being flash frozen [57].

To perform the analysis, the purified proteins FLAG-6xhis-G4P, 2G4P, FJG4P and 2FJG4P were labeled using Cy3 mono-reactive dye (Cytiva Life Sciences, USA). The labeling process involved incubating the proteins with the dye at 4 °C overnight in a Hepes-based buffer at pH 7.4. The reactive group of Cy3 mono-reactive dye is typically an N-hydroxysuccinimide (NHS) ester group, which selectively reacts with primary amine groups present in G4P and forms a stable covalent bond even under mild conditions. After the labeling process, the proteins were further purified using Heparin columns, and the labeling efficiency was determined by performing a full spectra scan using a spectrophotometer. Absorbance at 280 nm (for G4Ps) and 550 nm (for Cy3) were recorded. The concentrations of G4P and Cy3 were calculated using the extinction coefficients and Beer–Lambert law (C = A/(ε*l)), where C represents the concentration, A represents the absorbance, ε represents the extinction coefficient and l represents the light path length. Finally, the labeling efficiency was determined by dividing the concentration of Cy3 dye by the protein concentration. The calculated labelling efficiency for all the proteins was in the range of ~30% to 71%. 

### 4.4. Electrophoretic Mobility Shift Assay

Fluorescein labeled DNA oligos (Appendix A) were dissolved in a buffer containing 200 mM Tris at pH 7.4, 100 mM KCl and 1 mM EDTA to a final concentration of 2 µM. The mixture was then heated at 95 °C for 5 min to denature the DNA strands and slowly cooled down to allow for annealing and later diluted to 100 nM working concentration. The G4P proteins were incubated with DNA at 25 °C for 10 min followed by the resolving in 10% non-denature polyacrylamide gel and imaged using a ChemiDoc MP imaging system in fluorescein channel.

### 4.5. Mass Photometry

The MP experiments were conducted on Refyn Two-MP instrument (Refyn Ltd., Oxford, UK) on pre-cleaned coverslips (24 mm × 50 mm, Thorlabs Inc., Newton, NJ, USA) with serial washing with deionized water and isopropanol followed by drying. The silicon gaskets (Grace Bio-Labs, Bend, OR, USA) were cleaned in a similar process as coverslips and were placed onto coverslips for the experiments. The MP measurements were performed in an MP buffer containing 20 mM Tris pH 7.4, 100 mM KCl, 1 mM EDTA, and 10 mM MgCl_2_. The calibration was performed using a protein standard mixture: of β-amylase (Sigma-Aldrich, 56, 112, and 224 kDa, St. Louis, MO, USA), and thyroglobulin (Sigma-Aldrich, 670 kDa). Before each experiment, 15 μL buffer was placed into a chamber formed by coverslip-Gasket and focus was searched and followed by locking it using autofocus function. G4 DNA, G4P proteins or their mixtures were added to the chamber and mixed by pipetting. The movies were recorded for 60 s (6000 frames) using AcquireMP (Version 2.3.0; Refeyn Ltd., Oxford, UK) using standard settings. All movies were processed and analyzed using DiscoverMP (Version 2.3.0; Refeyn Ltd., Oxford, UK). Individual molecular weights readings for each experiment were binned into 3 kDa intervals, plotted as histograms and fitted to multiple Gaussians using GraphPad Prism. 

### 4.6. Single-Molecule Total Internal Reflection Microscopy

A custom-built prism TIRF microscope was used to perform single-molecule TIRF experiments. The microscope is built on an Olympus IX71 microscope frame and combines 532 nm (Compass 215M-50, Coherent Inc., Santa Clara, CA, USA) and 641 nm (Coherent, Cube 1150205/AD) laser beams using a polarizing beam splitting cube (CVI Melles Griot, PBSH-450-700-050), which are directed to the microscope objective at a 30° angle. TIR is achieved through a UV fused silica pellin–broca prism (325-1206, Eksma Optics, Vilnius, Lithuania) and an uncoated N-BK7 plano–convex lens (LA1213 Thorlabs Inc., Newton, NJ, USA). Photons are collected using a 60X, NA 1.20 water immersion objective (UPLSAPO60XW Olympus Corp., Shinjuku City, Tokyo, Japan), and spurious fluorescent signal is removed using a dual bandpass filter (FF01-577/690-25 Semrock Inc., Rochester, NY, USA). Cy3 and Cy5 emissions are separated using a dual-view housing (DV2 Photometrics, Tucson, AZ, USA) containing a 650 nm longpass filter (T650lpxr Chroma Technology Corp., Bellows Falls, VT, USA), and fluorescent images are captured using an Andor iXon 897 EMCCD (Oxford Instruments, Abingdon, UK).

### 4.7. Surface Tethered DNA Single-Molecule Experiments

Prior to surface tethering, the mixture biotinylated DNA oligo and indicated G4-forming oligos (see Appendix A) were heated together at 95 °C for 5 min and slowly cooled down to allow for annealing and later diluted to working concentration.

To extend the lifespan of fluorophores in single-molecule experiments, an oxygen scavenging system is necessary to reduce reactive oxygen species (ROS) that cause rapid photobleaching. We utilized 12 mM Trolox (6-hydroxy-2,5,7,8-tetramethylchroman-2-carboxylic acid) and Gloxy (catalase and glucose oxidase solution) to reduce ROS effects. Trolox is prepared by adding 60 mg of Trolox powder (238813-5G, Sigma-Aldrich) to 10 mL of water with 60 μL of 2 M NaOH, mixing for 3 days, filtering, and storing at 4 °C. Gloxy is prepared as a mixture of 4 mg/mL catalase (C40-500MG, Sigma-Aldrich) and 100 mg/mL glucose oxidase (G2133-50KU, Sigma-Aldrich) in wash buffer (25 mM Tris-HCl pH 7.0, 140 mM KCl); special KCl of Spectroscopy grade (#39795, Alfa Aesar, Haverhill, MA, USA) was used for all single molecule experiments.

The method for conducting single-molecule TIRF G4P-DNA binding experiments is as follows and includes all the DNA samples listed in Appendix A. Prior to the experiment, quartz slides (25 mm × 75 mm × 1 mm #1x3x1MM, G. Frinkenbeiner, Inc., Waltham, MA, USA) and cover glass (24 mm × 60 mm-1.5, Fisherbrand, Fisher Scientific, Hampton, NH, USA) are treated with passivation and PEGylation [60]. The flow cell is treated with 0.2 mg/mL NeutrAvidin (#3100, ThermoScientific, Thermo Fisher Scientific, Waltham, MA, USA) to tether biotinylated molecules to the flow cell surface. Excess neutravidin is removed with flow of 1mL wash buffer. To prepare the flow cell for imaging, 100 pM of biotinylated-DNA substrates are added and incubated for 3 min. Excess DNA is then removed using a wash buffer. For image collection, imaging buffer (containing 25 mM Tris-HCl pH 7.0, 140 mM KCl, 10 mM MgCl_2_, 1 mg/mL BSA, 1 mM DTT, 0.8% *w/v* D-glucose, 12 µM glucose oxidase, 0.04 mg/mL catalase and TROLOX) is added to the flow cell. The images are then collected using custom software, single.exe (generously provided by the Taekjip Ha Lab, JHU), with 532 nm laser power set to 45 mW. Image collection begins using 100 ms time resolution, gain of 290, background set to 400 and correction set to 1200. Once 300 frames have been collected, the range of concentration in pM to low nM of Cy3-G4Ps is added to the flow cell. Images are collected for a total of 1000 to 6000 frames (100 and or 300 s). 

### 4.8. Surface Tethered G4P Single-Molecule Experiments

Surface tethered G4P and its variants’ experiments are performed as stated for surface tethered DNA with the following exceptions. After removal of excess NeutrAvidin the flow cell is incubated with 150 pM biotinylated NTA beads (#90074, Biotium Inc., Fremont, CA, USA) for 3 min. Excess biotin-NTA is removed with wash buffer and various concentrations of each G4P variant were introduced in pM range followed by the incubation in the flow cell for 3 min. Excess G4P is removed with a wash buffer and an imaging buffer is added to the flow cell. An amount of 640 nm laser power is set to 45 mW and images are recorded as previously described. Once 300 frames have been collected the range of concentration of Cy5-DNA (DNA substrates were heated at 95 °C for 5 min and slowly cooled down to allow for annealing and later diluted to working concentration, see Appendix A) is added to the flow cell. The images were collected for a total of 1000 and/or 6000 frames (100 and/or 300 s). 

### 4.9. Single-Molecule Data Analysis

Fluorescent spot finding and trajectory extraction is done using an IDL script (generously provided by the Taekjip Ha Lab, JHU). Individual trajectories are then chosen for analysis using in-house MATLAB scripts (available upon request from the corresponding author). Trajectories were selected based upon the following criteria: no fluorescent intensity changes are present prior to frame 300 (when fluorescent molecules are added), baseline must be consistent throughout the trajectory and 2 fluorescent events persisting above the baseline for 3 frames must be present. After removal of the first 300 frames, the selected trajectories are then imported into hFRET [48] and fit to different states of fluorescent intensity. The best fit was determined from the largest log evidence lower bound comparison of the three tested models. Dwell times of transition state events are then extracted using KERA MATLAB software [49] and are subsequently plotted as a histogram. The histogram is then fit to a one- or two-phase exponential decay in GraphPad Prism. The best fit is determined from an F-test comparing the two exponential fits. The rate constant, k_−1_, is concentration independent and is determined from the best fit exponential of the histogram of the bound state dwell times. 

### 4.10. Single-Molecule FRET Experiments

Prior to smFRET experiments, the mixture of Cy3 and Cy5 labeled FRET oligos (see Appendix A) were heated at 95 °C for 5 min to denature the DNA strands and slowly cooled down to allow for annealing and later diluted to working concentrations.

The flow cells were treated with 0.2 mg/mL NeutrAvidin (#3100, ThermoScientific) for 3 min followed by the removal of excess Neutravidin by flowing 1 mL wash buffer (25 mM Tris-HCl pH 7.0, 140 mM KCl). To prepare the flow cell for imaging, 100 pM of DNA was added and incubated for 3 min followed by removal of excess unbound DNA by wash buffer. For image collection, imaging buffer (containing 25 mM Tris-HCl pH 7.0, 140 mM KCl, 10 mM MgCl_2_, 1 mg/mL BSA, 1 mM DTT, 0.8% *w/v* D-glucose, 12 µM glucose oxidase, 0.04 mg/mL catalase and TROLOX) was added and saturation amount of 200 nM G4P, 2G4P and 1 µM of FJG4P, 2FJG4P added and incubated. A total of 2-color fluorescence trajectories were extracted from short (100 frames) movies. FRET efficiency histograms were obtained by in-house MATLAB program using 10 different slide location data sets (each location contains 200–600 single molecules) for each condition. FRET efficiency was approximated as the ratio between the acceptor intensity and the sum of the acceptor and donor intensities after donor leakage correction to Cy5 channel. Calculated FRET values for each molecule in the selected movies were binned with the bin size of 0.01 and plotted and fitted to double Gaussian distributions using GraphPad Prism. 

## 5. Conclusions

In this study, we purified 4 variants of the G4-binding protein, G4P, and evaluated their binding to G4s with different topologies. This work reports the first single-molecule investigation of the G4P-G4 interaction. We found that 2G4P, which has four RSM elements, has higher affinity and broader recognition of both parallel and anti-parallel/mixed G4s than single G4P, perhaps due to a longer distance between RSM elements. We also observed that increasing the number of telomeric repeats enhanced the binding of all constructs. This may enhance the utility of these G4P variants in visualizing telomeric ssDNA. Single-molecule experiments revealed a complex interaction between 2G4P and 2hTelG4, which may involve binding to one or both G4 folds of the hTelG4 DNA. We also detected the formation of protein-G4 networks, especially with FJG4P constructs. Our findings contribute to the development of better tools for imaging and studying G4 DNA and its functions in cellular processes. The limitation of this study is that even improved G4P variants displayed fast dissociation kinetics. Both 2G4P and 2FJG4P can be further improved by enhancing stability of their G4-bound states. 

## Figures and Tables

**Figure 1 ijms-24-10274-f001:**
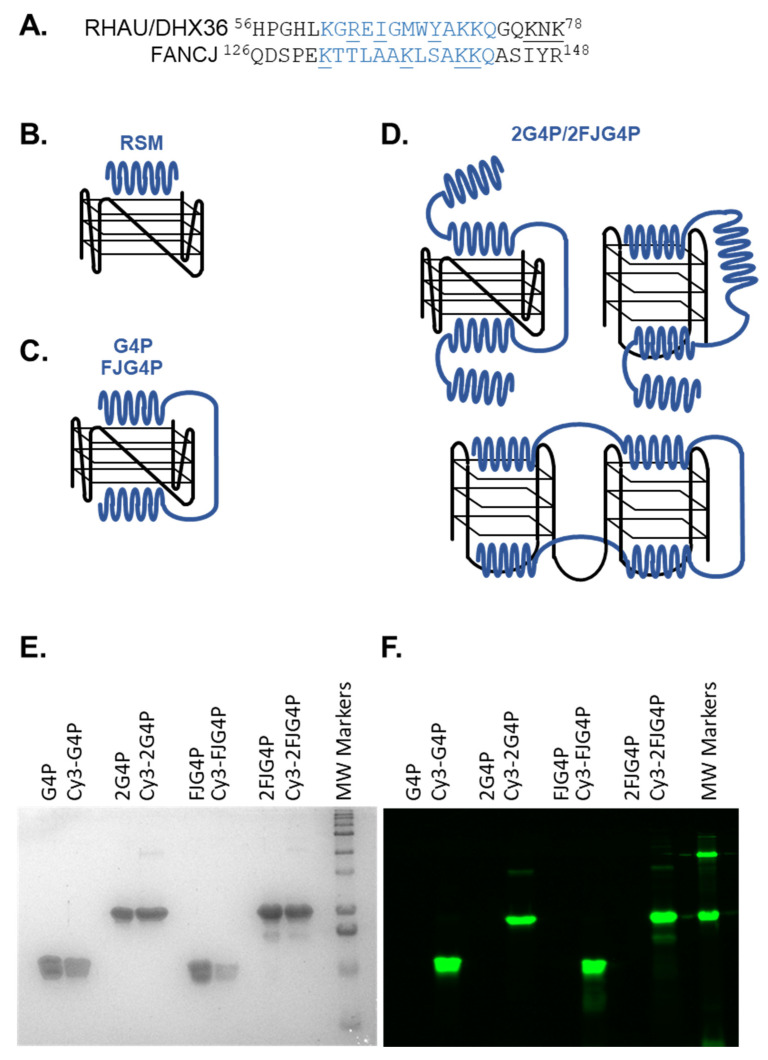
Design of the G4 sensors based on the G4P protein. (**A**) Alignment of the RHAU-specific motif and surrounding amino acids in human RHAU/DHX36 and FANCJ helicases. All residues whose involvement in G4 recognition has been experimentally confirmed are underlined. (**B**–**D**) Possible binding mechanisms of RSM, G4P, FJG4P and 2G4P/2FJG4P. (**B**) RSM peptide interacts with the terminal tetrade of G4. (**C**) Schematically depicts how the two RSM motifs in G4P may simultaneously bind the two terminal tetrads of the same parallel G4. The linker between the two RSM motifs may be too short to allow for similar interaction with a longer G4 in a hybrid or antiparallel conformation. (**D**) Doubling the number of G4 sensing elements could allow 2G4P binding to longer G4s and to multiple sequential G4s. (**E**) SDS-PAGE gel of purified G4P, 2G4P, FJG4P and 2FJG4P proteins labeled with Cy3 dye, followed by Coomassie staining. MW is a molecular weight marker. (**F**) Fluorescence scan of the same SDS-PAGE gel shown in (**E**) before Coomassie staining.

**Figure 2 ijms-24-10274-f002:**
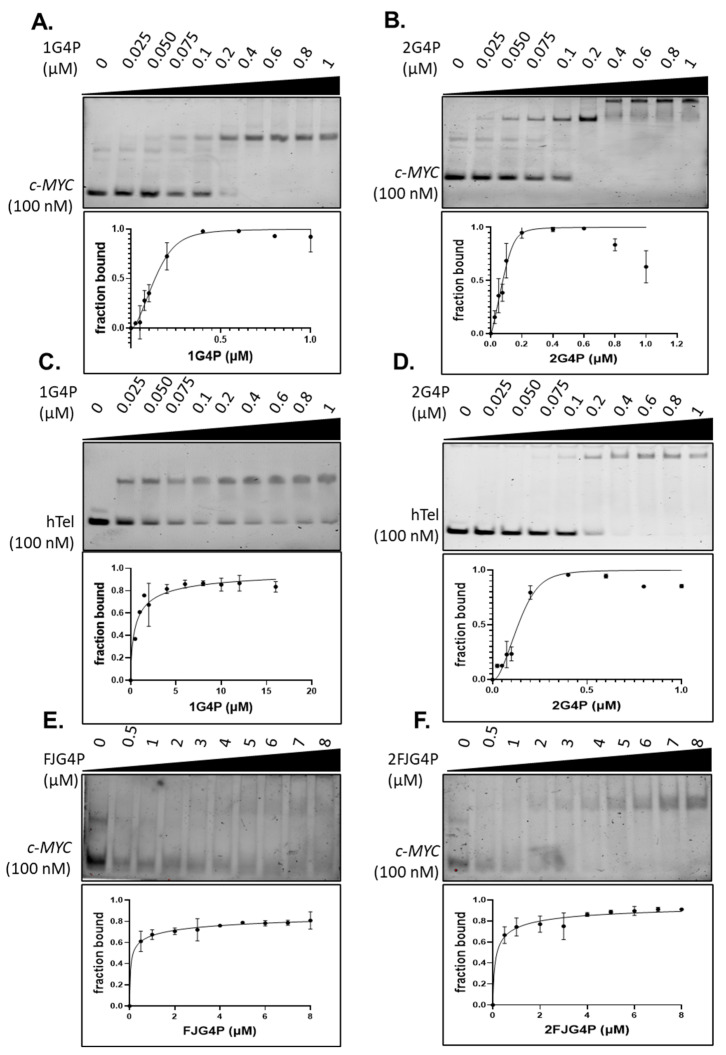
Electrophoretic Mobility Shift Assay (EMSA) demonstrating G4P’s stronger affinity for c-MYCG4 over hTelG4. (**A**) *G4P-c-MYC*G4, (**B**) 2*G4P-c-MYC*G4, (**C**) G4P-hTelG4, (**D**) 2G4P-hTelG4, (**E**) FJ*G4P-c-MYC*G4 and (**F**) 2FJ*G4P-c-MYC*G4.

**Figure 3 ijms-24-10274-f003:**
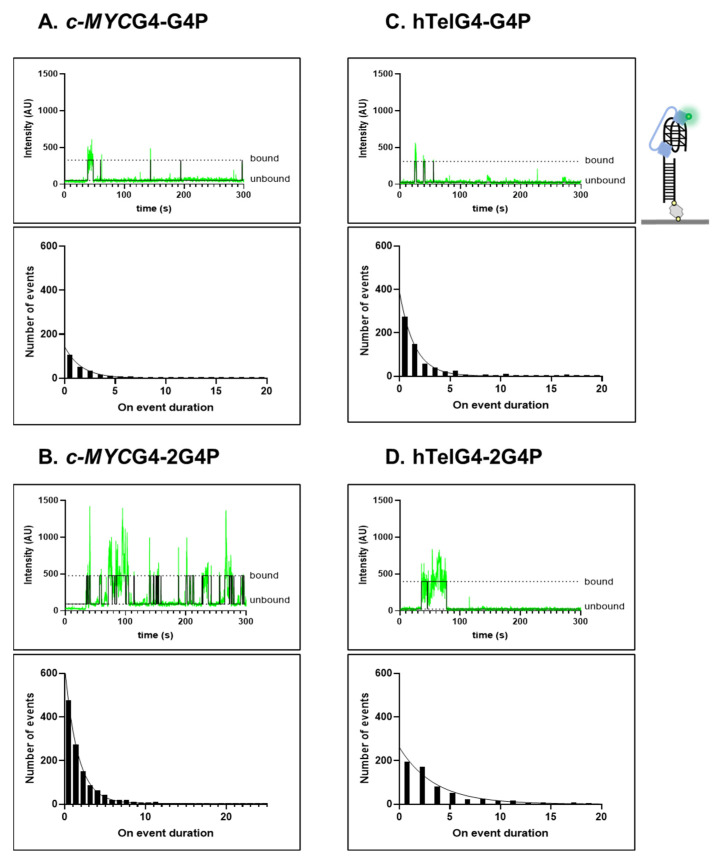
Single-molecule analysis of Cy3-labeled G4Ps binding to surface-tethered *c-MYC*G4 and hTelG4. DNA constructs containing biotinylated partial duplex DNA followed by a *c-MYC*G4 or hTelG4 were immobilized on a surface, while Cy3-G4Ps were introduced into the reaction chamber. For each construct, we show a representative fluorescence trajectory (green) overlaid with an idealized trajectory (black) and dwell-time distributions which were constructed from all “ON” dwell times and fitted using a single-exponential function: (**A**) *G4P-c-MYC*G4, (**B**) 2*G4P-c-MYC*G4, (**C**) G4P-hTelG4, and (**D**) 2G4P-hTelG4.

**Figure 4 ijms-24-10274-f004:**
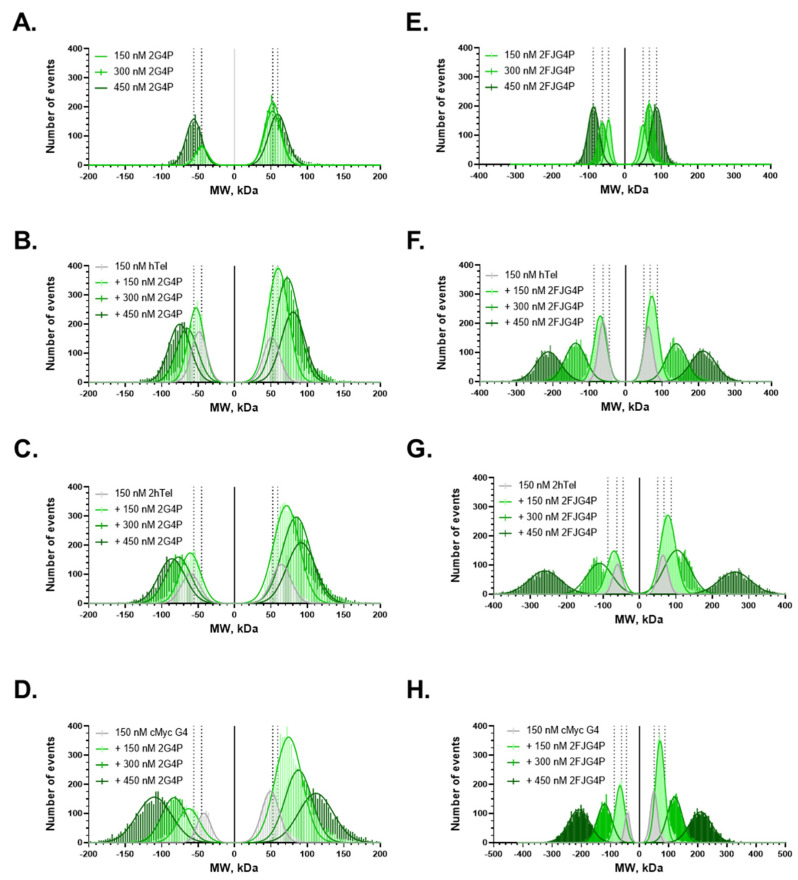
Mass photometry data. (**A**) shows the 2G4P alone at three different concentrations and (**B**–**D**) shows 2G4P with hTel, 2hTel and c-MYC. (**E**) shows the 2FANCJ at three different concentrations and (**F**–**H**) shows the 2FJG4P with hTel, 2hTel and c-MYC. The dotted vertical lines in each data set (**A**–**H**) can be used to visualize the shift in size compared to protein alone graph.

**Table 1 ijms-24-10274-t001:** Binding parameters derived from smTIRFM analyses.

	Protein-DNA	State 1 (k_on_) * s^−1^ M^−1^	State 2 (k_off_) s^−1^	State 2 (τ) s	Kd = k_off_/k_on_ (nM)
1.	***G4P-c-MYC*G4**	(2.16 ± 0.2) × 10^9^ s^−1^ M^−1^	0.61 ± 0.010	1.64 ± 0.03	0.22 ± 0.02
2.	**2*G4P-c-MYC*G4**	(3.9 ± 0.2) × 10^9^ s^−1^ M^−1^	0.62 ± 0.010	1.62 ± 0.03	0.16 ± 0.01
3.	**G4P-hTelG4**	(0.43 ± 0.036) × 10^9^ s^−1^ M^−1^	0.67 ± 0.015	1.5 ± 0.03	1.6 ± 0.1
4.	**2G4P-hTelG4**	(0.0845 ± 0.007) × 10^9^ s^−1^ M^−1^	0.29 ± 0.014	3.5 ± 0.2	3.4 ± 0.3
5.	**FJ*G4P-c-MYC*G4**	(3.45 ± 0.39) × 10^9^ s^−1^ M^−1^	0.94 ± 0.010	1.1 ± 0.01	0.27 ± 0.03
6.	**2FJ*G4P-c-MYC*G4**	(4.7 ± 0.26) × 10^9^ s^−1^ M^−1^	1.0 ± 0.023	1 ± 0.02	0.20 ± 0.01
7.	**FJG4P-hTelG4**	(0.01 ± 0.001428) × 10^9^ s^−1^ M^−1^	0.33 ± 0.011	3.1 ± 0.1	32.7 ± 4.8
8.	**2FJG4P-hTelG4**	(0.0339 ± 0.005357) × 10^9^ s^−1^ M^−1^	0.4 ± 0.008	2.8 ± 0.06	10.7 ± 1.7
9.	**G4P-2hTelG4**	(0.209 ± 0.1545) × 10^9^ s^−1^ M^−1^	0.4 ± 0.02	2.5 ± 0.1	1.9 ± 1.4
10.	**2G4P-2hTelG4** **(3 states)**	(0.0845 ± 0.00704) × 10^9^ s^−1^ M^−1^	NA	NA	NA
11.	**FJG4P-2hTelG4**	(0.0238 ± 0.003571) × 10^9^ s^−1^ M^−1^	0.35 ± 0.01	2.9 ± 0.09	14.5 ± 2.2
12.	**2FJG4P-2hTelG4**	(0.0642 ± 0.00714) × 10^9^ s^−1^ M^−1^	0.3 ± 0.01	3.35 ± 0.137	4.6 ± 0.55

* k_on_ calculations took into account the labeling efficiency of each protein.

## Data Availability

All data will be shared upon request.

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
