# Peer review of "Single-Molecule Analysis of the Improved Variants of the G-Quadruplex Recognition Protein G4P"

_ijms, 2023, doi:10.3390/ijms241210274_

Round 1
Reviewer 1 Report
The work is well conceived and well written, this area has been researched in great detail, but the results are significant and indicative: to get insight into the tandem repeats of G4P protein the following is found: G4P-G4 showed to outperform single G4P for binding to both cMycG4 (parallel G4) and hTelG4 (anti-parallel/mixed G4) and exhibit a preference for parallel topology over anti-parallel or mixed. 2G4P showed higher affinity of 2G4P for cMycG4 over G4P due to an increase in the association rate constant and no increase in the affinity of 2G4P for hTelG4. I suggest that the work be published in its entirety as written.
Author Response
We very much appreciate the reviewer's positive comments on our manuscript. Thank you!
Reviewer 2 Report
In the manuscript entitled “Single-molecule analysis of the improved variants of the G-quadruplex recognition protein G4P”, P. Gaur and coworkers investigated the affinity, selectivity and kinetic parameters of G4 binding to engineered small proteins, their tandem and hybrid derivatives based on the G4 recognition motifs of various DNA helicases. The strength of this study is the use of modern powerful experimental approaches, single-molecule total internal reflection fluorescence microscopy and mass photometry. The topic presented in this manuscript is interesting and relevant for deepening our knowledge of the factors affecting the stability, specificity, stoichiometry, and kinetics of protein binding to G4s of different topologies. Although the manuscript is well written and the results support the conclusions made by the authors, some issues need to be addressed before publication.
Major comments:
-
The trigger for this work was recent evidence that the prototype of protein used by the authors (G4P) specifically binds the G4 structures and displays better selectivity towards them than the previously published BG4 antibody; this small protein was also known to preferentially recognize parallel-stranded G4. However, the authors of the manuscript did not discuss the main objectives of this investigation. This could be improved protein tools for G4 visualization in living cells, or more ambitiously, understanding the general principles of G4 recognition by biologically important cellular proteins. It is known that most of the regulatory functions of G4s in the cell are mediated by the thermodynamic and kinetic (for example, sequestration of proteins on sporadically formed G4 structures within the genome regions) stability of specific G4-protein complexes.
-
Figure 1 shows the possible schemes for G4 targeting by engenired protein constructs. According to this Figure, the used tandem and chimeric protein ligands are able to interact only with terminal G-tetrads of G4 structures with different topologies. But in this case, the proteins used recognize mainly the quadruplex scaffold, and the marked preferential protein interaction with parallel G4s is ignored. Probably, other types of structural determinants should be involved in the specific recognition of G4s, for example, the space between two G4 subunits in telomeric multimeric G4s, quadruplex grooves, parameters of which differ for G4s of different topologies, phosphate groups, and loop bases. Please, give your comments.
-
It is difficult to accept that the authors aimed to develop less selective protein ligands, unable to discriminating G4 topologies, because the main trend in searching for the G4 ligands, at least small-molecule ones, is narrow specificity of such ligands.
-
It is necessary to add a Conclusions section, which will summarize the main goals and authors findings.
Minor comments:
The authors did not indicate whether they used human c-Myc, as a model of G4-forming sequence or not. If it is human one, it should be written in capital letters and in italics, like any genome fragment (c-MYC).
Author Response
Thank you for critically reading our manuscript and helpful suggestions.
Below are our point-by-point response to reviewer 2 comments. All the changes in the manuscript are easily trackable with track changes:
First comment: The trigger for this work was recent evidence that the prototype of protein used by the authors (G4P) specifically binds the G4 structures and displays better selectivity towards them than the previously published BG4 antibody; this small protein was also known to preferentially recognize parallel-stranded G4. However, the authors of the manuscript did not discuss the main objectives of this investigation. This could be improved protein tools for G4 visualization in living cells, or more ambitiously, understanding the general principles of G4 recognition by biologically important cellular proteins. It is known that most of the regulatory functions of G4s in the cell are mediated by the thermodynamic and kinetic (for example, sequestration of proteins on sporadically formed G4 structures within the genome regions) stability of specific G4-protein complexes.
Response: We have clarified the main objectives of the study in the Introduction section. We now state that the goal of the study was to develop protein ligands that can specifically and efficiently bind G-quadruplexes (G4s) irrespective of their topology. Another important outcome of our study is a mechanistic understanding of the tools (G4Ps) currently used to visualize G4s. (Page 3, line 94-98)
Second Comment: Figure 1 shows the possible schemes for G4 targeting by engenired protein constructs. According to this Figure, the used tandem and chimeric protein ligands are able to interact only with terminal G-tetrads of G4 structures with different topologies. But in this case, the proteins used recognize mainly the quadruplex scaffold, and the marked preferential protein interaction with parallel G4s is ignored. Probably, other types of structural determinants should be involved in the specific recognition of G4s, for example, the space between two G4 subunits in telomeric multimeric G4s, quadruplex grooves, parameters of which differ for G4s of different topologies, phosphate groups, and loop bases. Please, give your comments.
Response: We have updated figure legend to better explain our view on how G4P variants can interact with different G4s (Page 5, line 134-138). Briefly, the RSM motifs of RHAU and FANCJ helicases are expected to form short alpha-helices that interact with the top tetrad in the quadruplex. This is based on structural studies of the RSM peptide and RHAU helicase. The preference of the original G4P protein for parallel G4s, such as c-MYC may be due to two possible mechanisms. One mechanism is the compatibility between the RSM motif and the surface of the terminal tetrad. The other mechanism is the linker length between the two tetrads (Page 3-4, line 120 to 123). In Figure 1D we schematically depict interaction of 2G4P construct with “shorter” c-MYC G4 and longer telomeric G4. In the latter case, the increase length of the construct may allow binding of the two G4 sensors (RSM peptides) to two opposite terminal tetrads.
Third Comment: It is difficult to accept that the authors aimed to develop less selective protein ligands, unable to discriminating G4 topologies, because the main trend in searching for the G4 ligands, at least small-molecule ones, is narrow specificity of such ligands.
Response: We have clarified that the aim of the study was not to develop less selective protein ligands, but rather to develop ligands that are broadly applicable to detect various G4 structures irrespective of their topologies. Thus we aim for specificity towards G4s, but not a specific type of G4. This is indeed a different goal from that of small molecule stabilizers of a specific G4. The broad specificity reagents will be especially useful in visualization of telomeric G4s, which are poorly bound by the original G4P.
Fourth comment: It is necessary to add a Conclusions section, which will summarize the main goals and authors findings.
Response: We have added a Conclusions section that summarizes the main goals and findings of the study (Page 13, line 356-369).
Minor comments: The authors did not indicate whether they used human c-Myc, as a model of G4-forming sequence or not. If it is human one, it should be written in capital letters and in italics, like any genome fragment (c-MYC).
Response: We have corrected the spelling of "c-Myc" to "c-MYC" throughout the manuscript including supplementary information.
Round 2
Reviewer 2 Report
I am satisfied with the responses of the authors.